# Calf/female ratio and population dynamics of wild forest reindeer in relation to wolf and moose abundances in a managed European ecosystem

**Ilpo Kojola**[1]*, **Ville Hallikainen**[1], **Samuli Heikkinen**[2], **Jukka T. Forsman**[2], **Tuomas Kukko**[3], **Jyrki Pusenius**[4], **Paasivaara Antti**[2]

**1** Natural Resources Institute Finland (Luke), Ounasjoentie, Rovaniemi, Finland, **2** Natural Resources Institute Finland (Luke), Paavo Havaksentie, Oulu, Finland, **3** Natural Resources Institute Finland (Luke), Survontie, Jyväskylä, Finland, **4** Natural Resources Institute Finland (Luke), Yliopistonkatu, Joensuu, Finland

* ilpo.kojola@luke.fi

**Data Availability Statement:** Data are available at https://doi.org/10.5061/dryad.vhhmgqnvg.

## Abstract

### Background

The alternative prey hypothesis describes the mechanism for apparent competition whereby the mortality of the secondary prey species increases (and population size decreases decreases) by the increased predation by the shared predator if the population size of the primary prey decreases. Apparent competition is a process where the abundance of two co-existing prey species are negatively associated because they share a mutual predator, which negatively affects the abundance of both prey Here, we examined whether alternative prey and/or apparent competition hypothesis can explain the population dynamics and reproductive output of the secondary prey, wild forest reindeer (*Rangifer tarandus fennicus*) in Finland, in a predator-prey community in which moose (*Alces alces*) is the primary prey and the wolf (*Canis lupus*) is the generalist predator.

### Methods

We examined a 22-year time series (1996–2017) to determine how the population size and the calf/female ratio of wild forest reindeer in Eastern Finland were related to the abundances of wolf and moose. Only moose population size was regulated by hunting. Summer predation of wolves on reindeer focuses on calves. We used least squares regression (GLS) models (for handling autocorrelated error structures and resulting pseudo-$R^2$s) and generalized linear mixed (GLMs) models (for avoidance of negative predictions) to determine the relationships between abundances. We performed linear and general linear models for the calf/female ratio of reindeer.

### Results and synthesis

The trends in reindeer population size and moose abundance were almost identical: an increase during the first years and then a decrease until the last years of our study period.

**Funding:** The author(s) received no specific funding for this work.

**Competing interests:** The authors have declared that no competing interests exist.

Wolf population size in turn did not show long-term trends. Change in reindeer population size between consecutive winters was related positively to the calf/female ratio. The calf/female ratio was negatively related to wolf population size, but the reindeer population size was related to the wolf population only when moose abundance was entered as another independent variable. The wolf population was not related to moose abundance even though it is likely to consist the majority of the prey biomass. Because reindeer and moose populations were positively associated, our results seemed to support the alternative prey hypothesis more than the apparent competition hypothesis. However, these two hypotheses are not mutually exclusive and the primary mechanism is difficult to distinguish as the system is heavily managed by moose hunting. The recovery of wild forest reindeer in eastern Finland probably requires ecosystem management involving both habitat restoration and control of species abundances.

## Introduction

When sympatric prey populations share a common generalist predator their populations can be differentially related to the abundance of the predator. The two major theoretical concepts that predict the impact of the shared predator on its prey populations are the alternative prey hypothesis [1, 2] and the apparent competition hypothesis [3, 4]. The alternative prey hypothesis describes a mechanism where the predator functionally responds to the relative abundance of different prey species by shifting its diet between different prey species. The apparent competition hypothesis refers to the process where the predator population size positively responds to moose density that is likely to be the main determinant of prey biomass, and predicts that the consequent higher predator density increases more predation on the secondary prey.

Human-induced changes in the environment may differentially favor different prey species which may result in increase imbalances in predation on sympatric prey species. Habitat mediated apparent competition following long-term human-induced modifications is a complex task that has been rarely achieved for large mammals [5]. Well known example of such a suggested habitat change-mediated increase in predation rates is the declining population of threatened woodland caribou (*Rangifer tarandus caribou*) in North America [6–8]. Most woodland caribou populations are declining, and extirpation is ongoing [9–11]. The possible causes for the declines in caribou populations owing to predation are often based on the idea of apparent competition whereby anthropogenic changes in the environment favor other ungulate prey species, such as moose (*Alces alces*), which in turn boosts predator population growth and increase predation on caribou [5–9, 11, 12]. Calf recruitment rates in one woodland caribou population was negatively related to coyote (*Canis latrans*) abundance which was positively correlated with moose abundance [5].

The active control of species abundances might be a reasonable means to at least stop the decline in populations and provide more time for habitat restoration [8, 9, 13, 14]. Increasing the harvesting of predators' primary prey could decrease predator population size and thereby predation on caribou [9]. Combinations of treatments encompassing reductions in both predators and overabundant prey have produced the highest population growth rates [9]. Predator control is an option that might, however, be difficult to justify because large carnivores are iconic animals that often exist as small, threatened populations and are thought to provide ecosystem services [15, 16]. Ecosystem management plans aimed at the recovery of boreal wild

reindeer and caribou might require several concurrent actions to yield concrete results. Mature forests are key habitats for forest-dwelling *Rangifer* [12, 17–19], but habitat management alone might be insufficient because the restoration of key habitats may take too long to decrease the risk of extinction; thus, management plans relying only on habitat protection and restoration will likely fail [9].

In this study we examine the size of recently recolonized population of European wild forest reindeer (*R. t. fennicus*) in relation to adundances of grey wolf (*Canis lupus*) and wolves' primary prey, the moose. Ecology of wild forest reindeer is largely similar to that of woodland caribou [17]. Our study area resembles North American wolf-caribou ecosystem outside protected areas where logging is extensive and moose the primary prey of wolves. Wild forest reindeer were once distributed across the boreal coniferous zone in Europe but now are present only patchily, with a total population of approximately 10,000 animals, of which approximately only 2,300 exist in two populations in Finland [18, 20]. We have documented that the calf/female ratio in this particular population has formerly been strongly related to wolf population size using 10 years shorter time period [21].

We considered the relationships between wild reindeer, wolf and moose in the light of hypotheses of alternative prey [1, 2, 22–24] and apparent competition [4, 8, 25]. We assumed that predation by wolves influences the survival of reindeer calves in summer [21, 26, 27]. Whenever the alternative prey hypothesis accounts more we predict positive correlations of reindeer population size and the calf/female ratio in reindeer with the abundance of moose.

If the apparent competition hypothesis is the main driver, the abundance of the wolf population should be related positively to moose density which would bring about a decrease in the calf/female ratio in reindeer and, consequently, a decline in the reindeer population size. In our study area the moose population size is regulated by recreational hunting to limit browsing damage caused by moose. This may reduce apparent competition between reindeer and moose.

## Methods

### Study area

The study area is a ca. 6,000 km$^2$ area in east-central Finland (Fig 1). The region is dominated by highly managed productive boreal forests [28, 29]. Approximately 90% of the land area is covered by forests in which the main tree species are Scots pine *Pinus sylvestris*, Norway spruce *Picea abies* and two birch species (*Betula pubescens and Betula pendula*). The topography is flat, with the highest hills measuring 270 m above sea level. The terrain is characterized by lakes and peat bogs. Human settlements and high-traffic roads are scarce, but isolated houses and low-traffic roads are widespread in the study area. Other large carnivores that are known to kill wild reindeer are the brown bear (*Ursus arctos*), Eurasian lynx (*Lynx lynx*), and wolverine (*Gulo gulo*, A. Paasivaara unpublished data). The minimum brown bear population density in 2006 was estimated to be 16 bears/1000 km$^2$. Only wolf and brown bears prey on moose.

Wild forest reindeer populations rebounded our study area during the early 1960s after an absence lasting approximately 40 years [17]. The return of reindeer resulted from the expansion of reindeer populations from Russia. Wolves returned as permanent breeders in the mid-1990s [30]. Moose are the primary prey of gray wolves (*Canis lupus*) in European boreal forest ecosystems [29, 31, 32]. Moose population densities increased substantially during the 1970s largely due to widespread clearcutting, which created more habitats favored by moose [33]. In our study area, however, winter densities of the moose population were relatively low; in 2000–2015, the population densities of moose varied between 0.17 and 0.36 moose/km$^2$ (Pusenius et al., unpublished data). The moose population size was limited by harvest to control

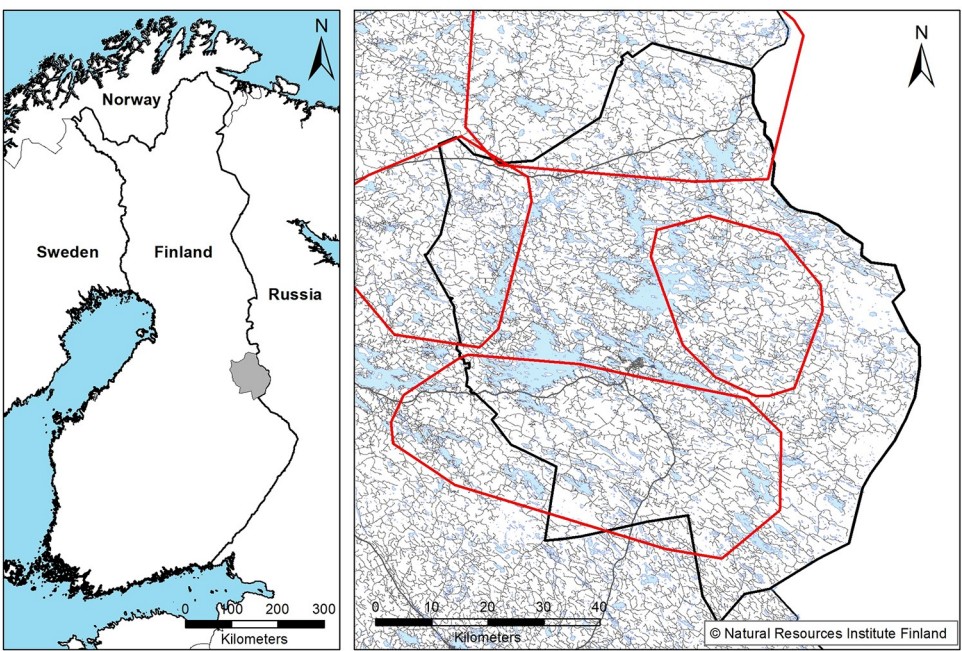

**Fig 1. Territory boundaries of GPS-collared wolves from winter 2010–2011, road network and the location the study area.**

browsing damage to forestry [34–36] and the number of traffic collisions [37]. Wild reindeer are protected from hunting. A few legal wolf removals have been conducted. Poaching has presumably had a large impact on the wolf population in eastern Finland [38].

## Data

We examined a 22-year time period (1996–2017). The population size of wild reindeer was assessed through yearly or biyearly total population counts by helicopter during late winter when reindeer are gathered up in their winter ranges [20]. For years when reindeer were not counted (1997,2002,2004), the population size was estimated as the mean of the population size before and after these years. Wolf population monitoring was based on a combination of snow-tracking of collared animals by experts, voluntary observations and genetic analyses [29, 30, 39]. Snow tracking was conducted to assess pack sizes. Wolf population estimates are for early winter. The moose abundance index is based on records by hunters. Moose hunting clubs are operating within their own territory of approximately 5 000 ha. Each club records the number of moose observations per hunting day during the autumn hunting season (from the last Saturday of September to the end of December) and the resultant index provides the annual mean number of moose seen per hunting day for our study area. We used this index because moose density was not estimated for years 1996–1999.

Numerical calf/female ratios of wild reindeer were based on field observations made by professional field technicians during September-November after the season of highest calf mortality, the first 80 days after the birth of the calf [21]. When examining relationships without lags, we related the calf/female ratio in reindeer to the previous winter's reindeer population size for density dependence, and to the wolf population estimate and moose abundance index for the same year. In models where the reindeer population size was an independent variable, we used the previous year's wolf population and moose abundance as independent variables, the time difference being 3–4 months with wolf and 3–5 months with moose.

## Statistical analysis

The data consisted of a time series of three wildlife populations in eastern Finland: population sizes for wild forest reindeer and wolves, and abundance index for moose. The aim of the study was to model the relationships between these populations without any lag effects between the abundances and with a one-year lag in the adundances. In addition, the relationship of the calf/female ratio to the wolf population was modeled with and without moose abundance as an independent variable. Interaction terms between wolf and moose abundances were tested to evaluate indication that wolf predation on reindeer might be related to moose abundance.

Strong correlations also existed between the abundances of reindeer and moose. Autocorrelations restricted the use of the modeling methods to only those methods where the autocorrelation could be estimated. The need to consider the autocorrelation was checked by using ACF plots and partial ACF plots, and by checking the residuals and how strongly they were correlated in time (ACF plots of the residuals). The Durbin-Watson test was used as an additional measure of autocorrelation of errors. The response variables were 1) reindeer (individuals), 2) wolves (individuals) and (3) the calf/female ratio in reindeer.

Population abundance models were performed using both generalized least squares (GLS) and generalized linear mixed (GLM) models. GLS could handle the auto-correlated error structures, and provide the reasonable pseudo-$R^2$ -values [40, 41]. On the other hand, Poisson models could not give the negative predictions otherwise to the GLS-models. Therefore prediction plots illustrating the effects of the explanatory variables were computed using the GLM-models [42].

The response variables in the models for the wild reindeer population size were log-transformed to normalize the distribution. The model for the wolf population did not require any log-transformation of the response variables; the transformation was found to weaken the distribution of residuals. Generalized least squares models provided AICs (Akaike Information Criterion [43]) that was used for comparing models with different autoregressive orders.

The need for testing autocorrelations was based on ANOVA using maximum likelihood estimation for alternative models. The final model was computed using restricted maximum likelihood (REML). The goal was to use AR orders (NULL, AR1, AR2) that would minimize AIC. In addition, the possible need for moving average parameters (MAs) was tested, but no MAs increased the model fit.

Because the response variables of the wildlife populations represented count data, Poisson or negative binomial models could have beeeen more appropriate to the responses. However, in R a generalized linear model (GLM, using R-function glm) without any random factor did not allow the use of autoregressive correlation structure for the error term, but the function glmmPQL did. The function glmmPQL with the Poisson family and an estimation the overdispersion were used in the "reference" (alternative) analysis for the GLS-models by building a "pseudo" random factor (one group). The variance of the "random effect" was computed (near zero) and the "residual" in the model output described the square-root of the dispersion parameter illustrating the overdispersion in the models. The results of these models are published in Table 2 in addition to the results of the GLS-models. The results with most of the models were close to the GLS results, and none of the interpretations of the results changed.

In some of the models, the temporal autocorrelation was so strong that the second-order autocorrelation was needed to obtain the sufficient standard errors and p values.The autocorrelated error ($\varepsilon_t$) can be described for the first-order auto-regressive process (AR(1)) as follows:

$$\varepsilon_t = \phi \varepsilon_{t-1} + v_t \qquad (1)$$

where the random shocks $v_t$ are assumed to be Gaussian white noise $\text{NID}(0, \sigma_v^2)$ and $\emptyset$ is the estimated first-order autoregressive coefficient between the two adjacent error terms.

The second-order autocorrelated error (AR(2)) can be described as follows:

$$\varepsilon_t = \phi_1 \varepsilon_{t-1} - \phi_2 \varepsilon_{t-2} + v_t \tag{2}$$

where $\emptyset_1$ and $\emptyset_2$ are the autoregressive coefficients for the first and second orders.

More information about the GLS regression and autocorrelated variance-covariance matrix was provided by Fox and Weisberg [40, 41]. The generalized linear Poisson models (estimated using PQL) were computed using the R statistical environment and the R package MASS [44]. These quasi-likelihood models did not provide AICs for model comparison. Generalized least squares (GLS) regression was computed using the R package nlme [45]. The beta-regression models with logit link function were computed and its function betareg [42]. Model predictions were computed using the R-package effects [46] The pseudo-R2 -values for the GLS models were computed using the R-package rcompanion. All the other computations were performed in the R statistical environment [47, R Core Team 2018].

The models for the calf/female ratio were performed using beta-regression which was the most reasonable choice to model the ratio. Betareg did not allow the use of autocorrelated error term in the model. However, the autocorrelation was not significant in the time series of the calf/female ratio (Durbin-Watson statistics = 1.754, p value = 0.203).

## Results

### Population trends

The wild forest reindeer population size increased from 1996 through 2001 and then rapidly decreased. The decrease became more moderate from 2008 onwards, but no signs of recovery existed (Fig 2). The trends in the moose abundance index were almost identical to the trends in reindeer population size but unlike the reindeer population size, they showed a decrease up to the end of the study period (Fig 2). Wolves returned by mid-1990, and their numbers increased rapidly from 1996–2001 but fluctuated from 2001 onwards (Fig 2). The reindeer population size and moose abundance index were more temporally autocorrelated than wolf population size (Fig 2).

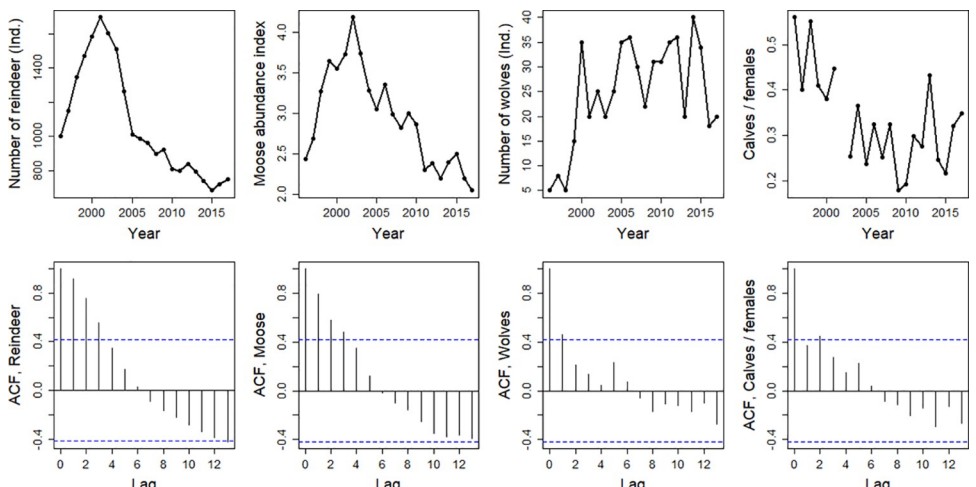

**Fig 2. Population sizes of wild forest reindeer and wolves, the abundance index of moose and calves/females ratio in reindeer with temporal autocorrelation functions in eastern Finland from 1996–2017.**

**Table 1. Student t-values and probabilities for generalized least squares (GLS) models and alternative GLMM-models (in parenthesis) for wild forest reindeer population size, the moose abundance index and the wolf population in eastern Finland from 1996–2017.** Based on residual autocorrelation tests, AR correlation structures were used in models if the residuals were autocorrelated. Cox & Snell pseudo $R^2$ were computed for the GLS-models.

| Dependent variable | Independent variable(s) | t | P | $R^2$ |
|---|---|---|---|---|
| Reindeer without lag (llog-normal gls) | Wolf population | -0.466 | 0.647 | 0.010 |
| | | (-0.803) | (0.431) | |
| Reindeer without lag | Wolf population | -3.584 | 0.002 | 0.878 |
| | | (-3.298) | (0.004) | |
| | Moose abundance | 10.934 | <0.001 | |
| | | (10.910) | (<0.001) | |
| Reindeer, one-year lag | Wolf population | 0.013 | 0.989 | -0.001 |
| | | (0.742) | (0.467) | |
| Reindeer, one-year lag | Wolf population | -3.863 | 0.001 | 0.782 |
| | | (-3.885) | (0.001) | |
| | Moose abundance | 6.703 | <0.001 | |
| | | (6.728) | (<0.001) | |
| Wolf without lag | Moose abundance | 0.744 | 0.466 | 0.003 |
| | | (0.584) | (0.566) | |
| Wolf without lag | Reindeer population | -0.505 | 0.619 | 0.000 |
| | | (-0.774) | (0.448) | |
| Wolf, one-year lag | Moose abundance | 0.023 | 0.982 | -0.011 |
| | | (0.063) | (0.950) | |
| Wolf, one-year lag | Reindeer population | -0.279 | 0.783 | -0.015 |
| | | (-0.475) | (0.640) | |

## Relationships between species abundances

**Reindeer.** Reindeer population size was not related to wolf abundance in models (GLSs or GLMs) where wolves were the only independent variable (Table 1), but models where wolf and moose populations were entered as independent variables, reindeer population size was negatively related to the wolf population size and positively related to the moose abundance (Fig 3). In models with a one-year lag, the reindeer population was not related to the wolf population alone but was related to both the wolf and moose populations in a model where both were entered as independent variables (Table 1). Akaike Information Criterion [39] could not be used for comparing models with one or two independent variables because of autoregression parameter in models with two independent variables and only pseudo R-squared figures could be calculated for the GLS models; however, based on the p values, the models with two independent variables appeared to fit better than the models with wolf population only (Table 1).

**Table 2. Statistics for the independent variables and the adjusted r-squared values in three beta regression models evaluating the relationship between the calf/female ratio, the wolf and reindeer population, and the moose abundance in eastern Finland from 1996–2017.**

| Independent variables | Estimate | Standard error | z | P | Pseudo- $R^2$ |
|---|---|---|---|---|---|
| Wolf population | -0.034 | 0.006 | -5.460 | < 0.001 | 0.565 |
| Wolf population | -0.031 | 0.006 | -4.930 | < 0 001 | |
| Reindeer population | 3.075e-4 | 2.156e-4 | 1.426 | 0.154 | 0.606 |
| Wolf population | -0.034 | 0.006 | -5.452 | < 0.001 | |
| Moose abundance | 0.042 | 0.123 | 0.341 | 0.733 | 0.568 |

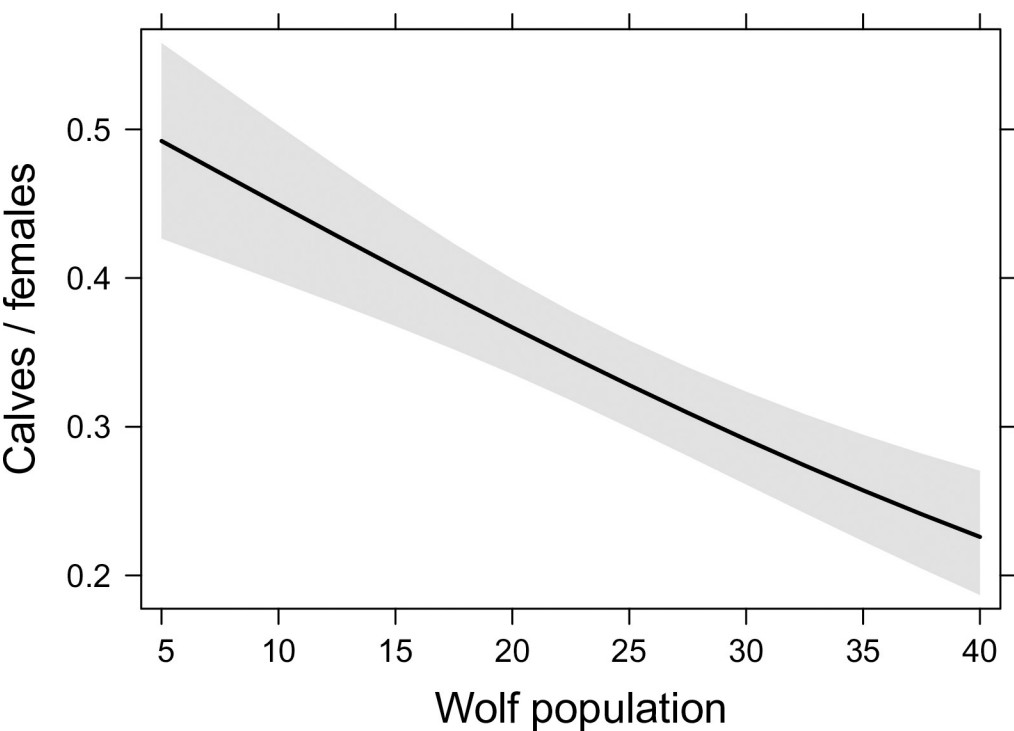

**Fig 3. Relationship of the calf/female ratio of wild forest reindeer to the wolf population in eastern Finland from 1996–2017 in a model where the wolf population was the only independent variable.**

**Wolf.**   In a model without a lag, the wolf population size was positively related to moose abundance and negatively related to reindeer population size (Table 1, Fig 4). In a model where the wolf population was the dependent variable with a one-year lag, neither the moose nor reindeer population was significantly related to the wolf population (Table 1). Reindeer population size was not significantly related to two-way interaction term wolf population size*moose abundance index (p value > 0.10).

**Calf/female ratios of wild forest reindeer.**   For reindeer, the yearly calf/female ratios from 1996–2001 were higher than those later in our study period (Fig 2). The annual growth rate of the wild forest reindeer population size (Y) was related to the calves/females ratio (X) in the linear model Y = -0.16 + 0.473 * X, t = 2.57, p = 0.020).

The beta-regression analyses showed that the calf/female ratio of reindeer was related to wolf population size in a highly significant fashion (Table 2, Fig 5). In a model where the population size of reindeer was entered as another independent variable for potential density dependence, the calf/female ratio was related to the wolf population size in a highly significant fashion, while no relationship between the ratio and reindeer population size existed (Table 2). Also in the model where moose abundance and wolf population size were entered as independent variables, the calf/female ratio was related only to wolf population size (Table 2). The calf/female ratio was not significantly related to the two-way interaction term wolf population size*moose (GLS; t = -0.697, p = 0.495, GLM; t = -0.856, p = 0.404).

## Discussion

Our main results provide some support to the alternative prey hypothesis but less to the apparent competition hypothesis. Population size estimates of the wild forest reindeer and moose

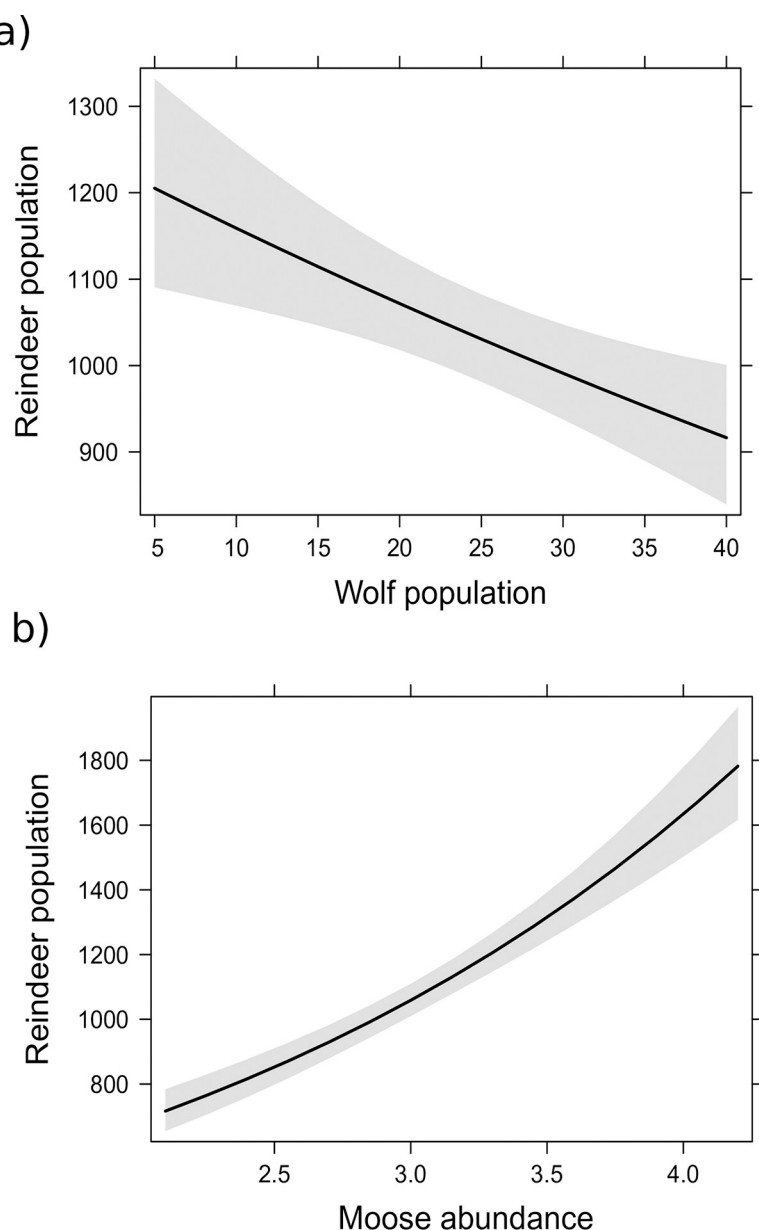

**Fig 4.** Relationships of wolf population (a) and moose abundance (b) to wild forest reindeer population in eastern Finland from 1996–2017 in models in which another sympatric ungulate was treated as another independent variable.

were generally positively associated and moose density was also positively associated with reindeer calf/female ratio. These results do not support apparent competition hypothesis. The simultaneous decrease of reindeer population size and moose abundance could be due to wolves' dietary shift from moose to reindeer when moose abundance was decreasing. However, our results must be interpreted cautiously as we examined only a part of the factors affecting the relative abundance of these species and our predator-prey system is heavily affected by humans, both through habitat changes and hunting on moose. The absence of two-tailed interaction wolf and moose abundances on reindeer population size and the calf/female ratio might indicate that moose abundance did not play a significant role for calf/female ratios in reindeer.

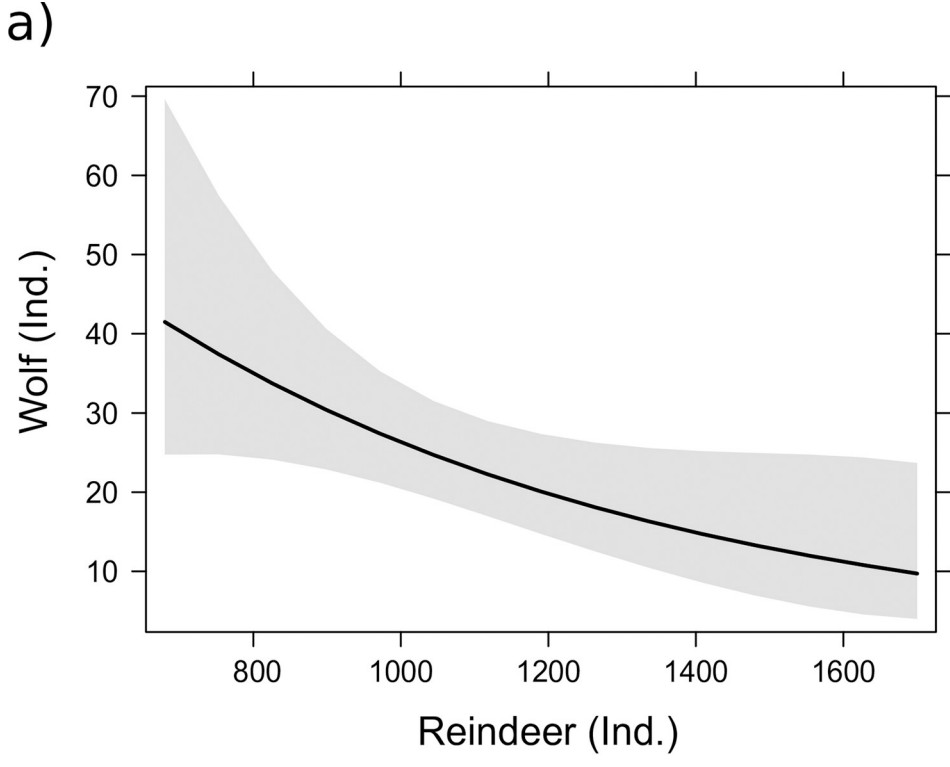

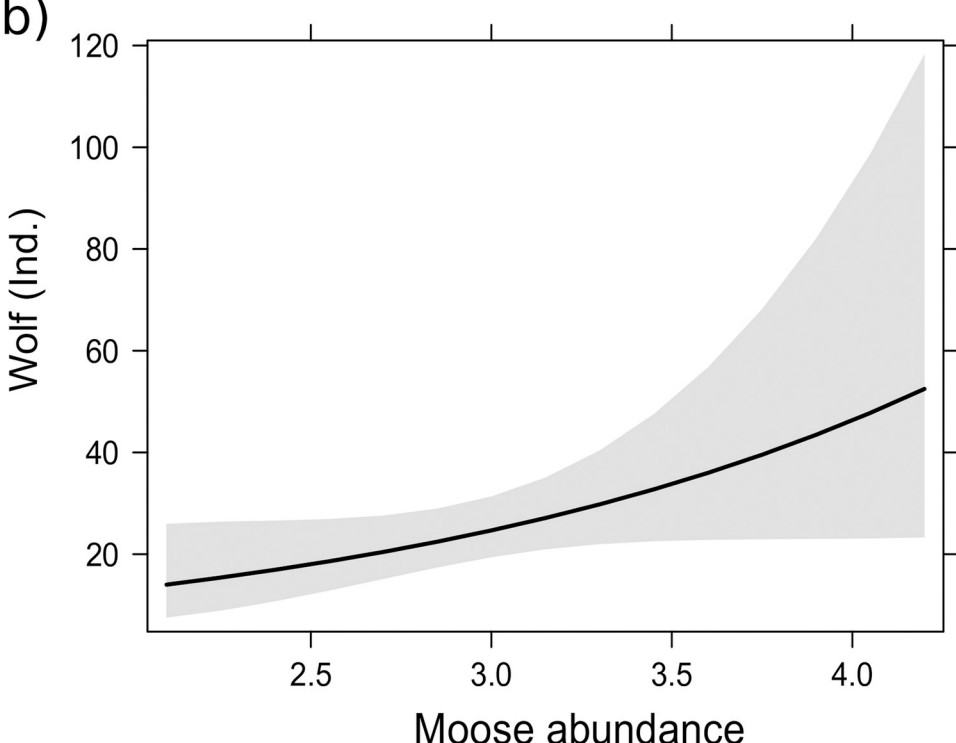

**Fig 5.** Relationships of wild forest reindeer population (a) and moose abundance (b) to wolf population in eastern Finland from 1996–2017 in models in which another sympatric ungulate was treated as another independent variable.

Vital rates may be related to diseases, parasites and weather conditions. We do not have data on diseases and parasites, but in 1996–2007 the calf/female ratio was not related to previous winters' snow depth or the timing of snowmelt in spring [20].

We found that calf/female ratio in reindeer was negatively related to wolf population size and the return of wolves could be one of the reasons that turned the reindeer population size trend from increase to decrease. Indeed, our previous studies have shown that in summer the most common kill of wolves calf reindeer [24]. Therefore the strong negative relationship of calf/female ratio to the size of wolf population most likely is due calf predation by wolves. Other large carnivores predate also on calves. However, in unpublished predation study from two GPS-collared bears tracked in our study area (methods for wolves [22]) within one month after the calving season of wild forest reindeer, only moose calves were found on kill sites. No data on predation by lynx and wolverine are available.

Reindeer population size was in close positive correlation with moose abundance but, owing to the correlative nature of our analyses we cannot draw conclusions about the reasons for this relationship. The stabilization of the reindeer population at the same time the moose population decreased to the lowest level might be in line with the relationships of woodland caribou populations to experimental reductions in moose populations in a Canadian mountain ecosystem where the decrease of caribou levelled off after moose abundances had decreased as a result of reduced apparent competition between caribou and moose [8]. However, in our study area wolf abundance was not related to moose abundance which is assumed by apparent competition hypothesis. A rapid decrease in the population of a principal prey species may cause a decline in the population of a secondary prey because predators may first consume more secondary prey, as suggested in the alternative prey hypothesis for cyclic populations [48–52]. A gradual decline in the primary prey is supposedly less detrimental to the secondary prey [25]. The similarities in the population dynamics between reindeer and moose and the stabilization of the reindeer population when the decline in the moose population slowly continued in our study area fit this assumption. However, predator-mediated apparent competition between prey species sharing a common predator would be most obvious when the predator population is responding to prey biomass, which is suggested to be reflected in encounter rates between the predator and the secondary prey [8, 53–55]. This response often occurs with a 1-2-year lag [25]. Our analysis did not provide significant evidence for the predator population's response to the moose abundance that is likely to be the primary determinant of prey biomass: the body mass of moose is about three times bigger than that of reindeer and densities in 2000–2015 were higher (0.17–0.36/km2, Pusenius unpublished data) than those of reindeer (0.13–0.28/km2).

The wolf population was remarkably labile in our study area. The wolf is officially a protected species in Finland. Some legal, lethal wolf removals occur for definite reasons, but all in all, known mortality is usually low and does not account for variation in population growth rates [38]. Population fluctuation was, instead, highly correlated with estimates of poaching based on the known and rumoured fates of GPS-collared wolves in eastern Finland [38].

The declines in the population of woodland caribou are largely connected to anthropogenic disturbances on interactions between caribou, predators and other prey species [5, 55]. Mumma et al. [56] reported that anthropogenic linear elements (roads and seismic lines) alone increased wolf predation on woodland caribou, but the authors did not find any relationships between anthropogenic elements, moose density and woodland caribou survival, although the predation risk was increased by caribou-moose co-occurrence. In our study area, disturbances are concentrated in a dense network of small roads constructed for forestry. These forest roads are preferred as travel routes by most wolf packs in eastern Finland [26]. Most of these roads were made before our study period, during 1970s and 1980s.

Population growth of North American woodland caribou was strongest where multiple recovery options (reductions of predators and overabundant prey, translocations, fenced refuges from predators) were applied simultaneously [9]. We assume that multiple actions should be included also in plans targeting the recovery of wild forest reindeer in Europe. A reintroduction into an almost predator-free region has resulted in a new population in central-Finland, but the growth rate of that particular population has been low [20], which emphasizes the need to protect habitats that are critical to boreal *Rangifer* [12, 57]. Caribou mortality is higher in disturbed than undisturbed landscapes [12, 55]. Restoration of key habitats would require such large changes in forest management strategies within such wide areas that such plans may remain unrealistic. The ecology of forest-dwelling *Rangifer* is relatively similar across the entire coniferous zone, and in both ecosystems of North America and Finland, these deer may periodically be dependent on arboreal lichens due to deep snow that precludes cratering for terrestrial lichens [17, 58], the latter of which is the main winter forage of wild forest reindeer in eastern Finland [19], and e.g., in montane ecosystems in Alberta [59]. The biomasses of arboreal lichens are much higher in old-growth forests than in managed second-growth forests [60, 61]. The biomasses of ground lichens are also highest in old-growth forests, but their relationship to stand age is not as clear as that of arboreal lichens [61, 62]. Sustainable recovery of wild forest reindeer, however, probably requires ecosystem management where one component is habitat management. In the human-modified forest landscape, active control of the species abundances appears to be necessary for the population recovery of *Rangifer*. The reduction in the abundance of primary prey might decrease the predation risk although this was not supported in our study. Potential plans for the removal of predators should take into account the viability of predator populations. For example, in Finland, the brown bear and lynx are not threatened species, unlike the wolf, which is highly endangered nationally [63]. To control predation by bears and lynx, regional license allocation for leisure hunting, which is the primary method for regulating bear and lynx abundances in Finland [21], could constantly inform the vulnerability of wild forest reindeer within the coming decades. However, the managing of wild forest reindeer population should preferably be cautious whenever the mechanism is not identified [64]. For example, habitat mediated apparent competition appears to decouple in northernmost ranges of woodland caribou where moose and wolf densities are low [64] likewise in our study area.

## Author Contributions

**Conceptualization:** Ilpo Kojola, Samuli Heikkinen, Jyrki Pusenius, Paasivaara Antti.

**Data curation:** Tuomas Kukko, Jyrki Pusenius.

**Formal analysis:** Ville Hallikainen.

**Methodology:** Tuomas Kukko, Paasivaara Antti.

**Project administration:** Jukka T. Forsman.

**Resources:** Jukka T. Forsman.

**Validation:** Tuomas Kukko.

**Writing – original draft:** Ilpo Kojola.

**Writing – review & editing:** Ilpo Kojola, Jukka T. Forsman, Tuomas Kukko, Jyrki Pusenius, Paasivaara Antti.

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
