## [Decision Letter · Decision Letter 0]

13 May 2021

PONE-D-21-02662

Predation by wolves on European wild reindeer in a managed boreal ecosystem

PLOS ONE

Dear Dr. Kojola,

Thank you for submitting your manuscript to PLOS ONE. After careful consideration, we feel that it has merit but does not fully meet PLOS ONE’s publication criteria as it currently stands. Therefore, we invite you to submit a revised version of the manuscript that addresses the points raised during the review process.

The submitted manuscript is important as it provides useful information on prey and predator interaction. However, there are critical issues highlighting by the reviewers which needs to be addressed in the revised version. 

We look forward to receiving your revised manuscript.

Kind regards,

Lalit Kumar Sharma, Ph.D

Academic Editor

PLOS ONE

Additional Editor Comments:

In addition to the comments of two reviewers I also feel that there are issues related to hypothesis tested to understand the interaction of wolf and reindeer. Both of the reviewers have significantly improved the manuscript in terms of language and readability. Hence, the authors should address the comments of both the author in the revised manuscript.

Journal Requirements:

3. We note that Figure 1 in your submission contain map images which may be copyrighted. All PLOS content is published under the Creative Commons Attribution License (CC BY 4.0), which means that the manuscript, images, and Supporting Information files will be freely available online, and any third party is permitted to access, download, copy, distribute, and use these materials in any way, even commercially, with proper attribution. For these reasons, we cannot publish previously copyrighted maps or satellite images created using proprietary data, such as Google software (Google Maps, Street View, and Earth). For more information, see our copyright guidelines: http://journals.plos.org/plosone/s/licenses-and-copyright.

3.1.    You may seek permission from the original copyright holder of Figure(s) [#] to publish the content specifically under the CC BY 4.0 license. 

3.2.    If you are unable to obtain permission from the original copyright holder to publish these figures under the CC BY 4.0 license or if the copyright holder’s requirements are incompatible with the CC BY 4.0 license, please either i) remove the figure or ii) supply a replacement figure that complies with the CC BY 4.0 license. Please check copyright information on all replacement figures and update the figure caption with source information. If applicable, please specify in the figure caption text when a figure is similar but not identical to the original image and is therefore for illustrative purposes only.

Reviewers' comments:

Reviewer's Responses to Questions

**Comments to the Author**

1. Is the manuscript technically sound, and do the data support the conclusions?

Reviewer #1: Partly

Reviewer #2: Yes

2. Has the statistical analysis been performed appropriately and rigorously? 

Reviewer #1: Yes

Reviewer #2: Yes

3. Have the authors made all data underlying the findings in their manuscript fully available?

Reviewer #1: No

Reviewer #2: No

4. Is the manuscript presented in an intelligible fashion and written in standard English?

Reviewer #1: Yes

Reviewer #2: Yes

5. Review Comments to the Author

Reviewer #1: The authors present four time series 1996-2017 from eastern Finland: 1) forest reindeer abundance, 2) forest reindeer calf/female ratio, 3) moose density index and 4) wolf abundance. They aim at testing the alternative prey hypothesis and the theory of apparent competition, with the reindeer being the alternative prey and moose the main prey of wolves. Moose are regulated by humans, while reindeer and wolves are not (except of some legal culling and unquantified poaching of wolves). They do not present data on predation itself, and so they test for numerical responses of predator and prey. In many ways, the system is similar to that of the woodland caribou in N-America, where human landscape modification (logging, seismic lines) increased wolves’ main prey, and where research supports apparent competition between woodland caribou and other deer species. The same authors have published a study on the reindeer-moose-wolf-system in eastern Finland in 2009, but now they present ten more years of data.

General comments:

1) Although you merely present time series and test for correlations between them, you make very strong conclusions on causal relationships (apparent competition and alternative prey hypothesis moose-reindeer-wolf). As you mention in the discussion, there are many other factors that could contribute to the observed population dynamics in forest reindeer, such as bear and lynx population, gravel roads, lichen abundance, and there might be more (disease and parasites? Snow and icing conditions?). Still, you “only” present data on wolf and moose abundance as predictors of reindeer abundance and calf/female ratio. This might be because you do not have access to the other time series. Anyway, or even more importantly, you need to make this clear to the reader in the introduction already, that you only look into one of several limiting factors, and you need to reword your text throughout the manuscript to make it less “causal”.

2) Alternative prey hypothesis APH and apparent competition ACH: You want to test if any of these two hypotheses may be supported in your system. I propose you introduce the reader better to how these two processes may affect the reindeer population. The APH is in the first hand a functional response of the predator to changes in the availability of its main prey, i.e. switching to alternative prey when main prey abundance is low. The functional response of the predator will have an effect on the mortality rate of the alternative prey and therefore change population structure (in case of age-specific mortality) and finally population growth. You would therefore exoect a positive relationship between moose and reindeer densities. The ACH on the other side describes the numerical response of the predator to changes in main prey density, which then spills over to predation rates in the alternative prey. You would expect a negative relationship between reindeer and moose abundance. These two hypotheses are not mutually exclusive, so both may take place in the same system. As a result, the positive effect from APH and the negative effect from ACH would be dampened. It could be informative to show this graphically in the introduction, and this could lead you to make more clear predictions for the reindeer population and calf/female ratio in the end of the introduction.

3) Your analyses 1: it is difficult to follow your motivations for the different models. Probably the wolf is predating mostly on the juvenile segment of the reindeer, that is why you test the calf/female ratio ~ wolf abundance. Make this more clear in the introduction already.

4) Your analyses 2: You first tested reindeer abundance ~ moose abundance and found a strong positive relationship between the two, with and without lag. Still, you chose to include both reindeer and moose abundance as predictors when modelling a) the calf/female ratio and b) wolf abundance. The strong collinearity of reindeer and moose abundance can give faulty model results and should be avoided.

5) Your results and discussion: Your main results are 1) negative relationship for calf/female ~ wolf abundance, which might indicate calf predation by wolves; 2) a negative relationship for wolf and positive for moose for reindeer abundance ~ wolf abundance + moose abundance; 3) No relationship between wolf abundance ~ moose abundance. Although the latter two findings give some support to APH, but not to ACH, you still put a lot of emphasis on ACH in the discussion, and you even propose reduction of moose population size as a measure to decrease predation on reindeer. I think evidence for this statement is not present in your results.

6) The English needs to be improved. I have not focused on he language in my review, but can do so in a revised manuscript.

Specific comments:

L1: The title is misleading. You did not study predation. You assume that predation may be an underlying mechanism of the observed correlations between the time series, that’s all.

L10-13: rewrite this first paragraph of the abstract. Include the alternative prey hypothesis and theory of apparent competition, end up with your predictions – see general comments

L39-40: the message of this sentence is unclear as it is written know. You refer to the paper of Holt and Lawton (1994) and took some of their wording in the abstract into your sentence. They write “…Theory suggests victim-species coexistence depends on particular conditions…”. By using the same wording, the context went missing. I think you can delete this first sentence and rewrite the entire first paragraph of the introduction. Explain the two hypotheses that describe direct and indirect interactions of prey species with a shared predator.

L41: Apparent competition – is it a theory or a hypothesis? I see both is used in literature. I personally would prefer hypothesis, so it is on the same line as the alternative prey hypothesis.

L42: I would put APH before ACH, because APH describes a mechanism (prey switching), while ACH is more of a process. This would also change the order in the text of the following lines.

L68-73: It would make sense to first talk about predator control, so move the last sentence up, and so talk about the regulation of the main prey, and so the combination of both.

L86: The reproductive rate is so much more than calf/female ratio. Make it more specific and introduce the reader to the wolf being a calf specialist – if this is true.

L103: Any chance to incorporate bear data into your models? I think they might be an important predator on neonate calves. What about lynx?

L124: Why had collared animals to be snow-tracked? To find pack size? Clarify!

L126-127: Clarify!

L132: So the reindeer counts were a sample, not a full count? Or is the number of observations identical with abundance?

L133: Number of observations: Is this the number of females, the number of animals, or the number of days people have been out observing?

L138: I guess this needs to be dependent, not independent.

L160 onwards: GLM with proportional data, GLS and AR-structures are standard methods, and there is no need to show the model equations. However, the calf/female ratio is not really a proportion, it is a ratio. For this, beta-regression would be more adequate than a GLM with binomial family.

Results: I would move the abundance sub-chapter (relationships between species-abundances) up, so it comes right after the first paragraph on trends.

L243: Where do you present those pseudo R-squared?

L248: See general comment on collinear predictors, so I don’t think you can include both reindeer and moose. But anyhow: What do you define as significant? In Table 2, p-values of those two predictors are > 0.05.

L252: No, you did not study predation on boral wild reindeer! See general comment 1).

L255: …major reason…? Rather: …one of the reasons…

L261-264: But the wolf population did not really decrease after moose population decreased. Which brings to my mind that the ACH should test predator numerical response not only as a function of its principal prey, but rather of the total prey base, i.e. the sum of available prey biomass irrespective of species. You however use an index for moose abundance, so it will be difficult to estimate the sum of available biomass of reindeer and moose.

L288: I assume the gravel roads are a consequence of forestry. Clarify, maybe call them forest roads. Has their density changed throughout the years? Or has their use changed?

L292-293: This is a really strong statement that has no direct support from your data, see also general comments 1) and 5).

L314: Moose abundance was positively related with reindeer abundance, and I consider the model looking into calf/female ratio ~moose abundance + reindeer abundance as invalid due to collinearity of the predictors. So no support for a negative relationship between moose abundance and “predation risk”.

L525 and else in the text: You use abundance, population, population size and density (e.g. Figure 5b) intermittently. Be consistent.

Figure 1: Finland is not labelled, while all other countries are! The map allows for inclusion of more information, which could make the study easier to understand. You could for example add wolf territories, e.g. as overlapping territories from all years. And/or the network of forest roads. Maybe also the area of the other wild reindeer population. Anything to make it more informative!

Figure 2: It would make sense to include two similar graphs for calf/female ratio.

Figure 5: The y-axis goes below zero, which does not make sense for wolf abundance. This makes me think that you maybe rather use a Poisson-regression when comparing wolf abundance (a count) to moose abundance index (or reindeer abundance).

Reviewer #2: The authors have been studying wolf predation on wild forest reindeer in a system where an endangered predator (wolf) has a primary non-endangered prey species (moose) and secondary endangered prey species (wild forest reindeer). In the manuscript the authors have used correlative approach to investigate species abundance interactions on reindeer and wolf population sizes. In addition, they have also used the same approach to investigate calf/female ratio in winter herds of wild forest reindeer. In former studies this ratio has been shown to be related on wolf abundance and in this manuscript the authors have broadened their former study by analysing longer data period and adding the yearly abundance of moose into their models. Their main findings were that the calf/female ratio was negatively associated negatively to both wolf population size and moose abundance. Wild forest reindeer size was only dependent on wolf population size in model where the moose abundance was entered as another independent variable. There was not strong evidence on effect of moose abundance or reindeer population size on wolf population size.

The long term time series for wildlife species used in the study are imposing and the statistical analyses are rather comprehensive, even though there have been some problems in the analyses because of autocorrelations. The study system and the results are very interesting. However, the interpretation of the causal effects behind the species population size variation are difficult because of the correlative approach used in the study. The fact that moose population size is heavily regulated via hunting and wolf population is fluctuating mainly because of poaching makes interpretation of the results even more difficult. However, the observation that the predation by wolves on reindeer might be influenced by moose abundance could still have substantial management implications and the authors are discussing praiseworthy on possible management actions.

The paper is generally well written. However, especially in the abstract the authors should present their results more comprehensively to make the message of the paper clear (see below). And, please, do not use “population” as a substitute of “population size”.

Minor points:

Abstract:

Line 15: “…reproductive rate…”. What does this mean? Is it your calf/female ratio. Reproduction is generally measured by gross reproduction rates or net reproduction rates that generally indicate the ratio between the sizes of the daughter's and mother's generations and you are not really measuring them.

Line 15: “wild reindeer”. Sometimes wild reindeer and wild forest reindeer. Please, be consistent.

Line 20: “Reindeer and moose abundances were highly correlated…”. Is this already a result and should be in your Results section of your abstract. You are actually repeating this result also on line 24-25. “The trends in reindeer population size and moose abundance were almost identical”.

Line 27: “Change in reindeer population between consecutive winters”. Should not this be “Change in reindeer population *size* between consecutive winters”.

Line 28: “The calf/female ratio was closely related to wolf population size”. The calf/female ratio was closely related *negatively* to wolf population size.

Line 29: “the reindeer population was related to the wolf population”. I suppose that you mean “the reindeer population *size* was related *negatively” to the wolf population *size*. Other wise this text sounds funny; and tell also that the relationship is negative. …”

Line 30: “The wolf population was…”. Should be “The wolf population *size* was

Introduction: “

Lines 85-90: I suggest that the authors construct a table where they show what are the predictions of their two theoretical models and how their different results support or do not support the two hypotheses.

Data:

Lines 132-133: “the calf/female ratio was weighed against the annual number of observations”. In the analyses?

Lines 145-146: “The aim of the study was to model the relationships between these populations”. Or: “The aim of the study was to model the interactions in abundances of these populations” or did you really model the relationship between the populations?

Line 147-148: “In addition, the relationship of the calf/female ratio to the wolf population..” Or: “In addition, the relationship of the calf/female ratio to the wolf population *size*”.

Result:

Line 217: “Reindeer population and moose abundance.”Or “ Reindeer population *size* and moose abundance “

Line 221: “The annual growth rate of the wild forest reindeer population was related to the number of calves weighted by the number of females (the calves/females ratio) in the linear model Y = -0.16 + 0.473 * X, t = 2.57, p = 0.020).” Is the number of calves weighted by the number of females” really correct here. Thus what are your Y and X here. Please, clarify.

Line 225: “but the ratio was related to the wolf population” …. Should be: “but the ratio was related to the wolf population *size*”.

Line 227-228: “the calf/female ratio was related to the wolf population in a highly 28 significant fashion”. Should be: “the calf/female ratio was related to the wolf population *size*” in a highly significant fashion”.

Line 228-229: “while no relationship between the ratio and the reindeer population *size* existed”. However, there was significant association between reindeer population size and calf/female ratio in your model with three independent variables. Should you mention it, too?

Line 237-241: “but in a model where wolf and moose population *sizes* were entered as independent variables, reindeer population size was negatively related to the wolf population *size* and positively related to the moose population *size*. In models with a one-year lag, the reindeer population *size* was not related to the wolf population *size* alone but was related to both the wolf population *size* and moose*abundance* in a model where both were entered as independent variables”.

Lines 247-250: “Wolf. In a model without a lag, the wolf population size was positively related to moose abundance and negatively related to reindeer population size (Table 2, Fig. 5). In a model where 249 the wolf population size was the dependent variable with a one-year lag, neither the moose nor reindeer population *size* was significantly related to the wolf population (Table 2).

Discussion:

Lines 254-255: “We found that calf/female ratio in reindeer was negatively related to wolf population *size*…

Lines 260-261: “The stabilization of the reindeer population size at the same time the moose population *abundance* decreased…”

Figures:

Fig 4: “Relationship of the calf/female ratio of wild forest reindeer to the wolf population *size*in…”

Fig 5: “Relationships of reindeer population *size*(a) and moose abundance (b) to wolf population *size*…”

6. PLOS authors have the option to publish the peer review history of their article (what does this mean?). If published, this will include your full peer review and any attached files.

Reviewer #1: No

Reviewer #2: No

---

## [Author Response · Author response to Decision Letter 0]

28 Sep 2021

Responses to referees’ comments:

REFEREE 1: General comments:

1) Although you merely present time series and test for correlations between them, you make very strong conclusions on causal relationships (apparent competition and alternative prey hypothesis moose-reindeer-wolf). As you mention in the discussion, there are many other factors that could contribute to the observed population dynamics in forest reindeer, such as bear and lynx population, gravel roads, lichen abundance, and there might be more (disease and parasites? Snow and icing conditions?). Still, you “only” present data on wolf and moose abundance as predictors of reindeer abundance and calf/female ratio. This might be because you do not have access to the other time series. Anyway, or even more importantly, you need to make this clear to the reader in the introduction already, that you only look into one of several limiting factors, and you need to reword your text throughout the manuscript to make it less “causal”.

RESPONSE: We have made extensive changes into Abstract, Introduction and Discussion. We reworded the also the title so that causal relationships between reindeer, moose and wolf are not assumed. We do not have data on diseases and parasites which is now notified in Discussion. We refer to one former study where no relationship of the calf/female ratio to snow depth and the timing of snowmelt was not detected in this reindeer population. 

2) Alternative prey hypothesis APH and apparent competition ACH: You want to test if any of these two hypotheses may be supported in your system. I propose you introduce the reader better to how these two processes may affect the reindeer population. The APH is in the first hand a functional response of the predator to changes in the availability of its main prey, i.e. switching to alternative prey when main prey abundance is low. The functional response of the predator will have an effect on the mortality rate of the alternative prey and therefore change population structure (in case of age-specific mortality) and finally population growth. You would therefore expect a positive relationship between moose and reindeer densities. The ACH on the other side describes the numerical response of the predator to changes in main prey density, which then spills over to predation rates in the alternative prey. You would expect a negative relationship between reindeer and moose abundance. These two hypotheses are not mutually exclusive, so both may take place in the same system. As a result, the positive effect from APH and the negative effect from ACH would be dampened. It could be informative to show this graphically in the introduction, and this could lead you to make more clear predictions for the reindeer population and calf/female ratio in the end of the introduction.

RESPONSE: We clarified the predictions for our study population based on hypotheses of alternative prey and apparent competition (in this order as suggested by the referee). 

3) Your analyses 1: it is difficult to follow your motivations for the different models. Probably the wolf is predating mostly on the juvenile segment of the reindeer, that is why you test the calf/female ratio ~ wolf abundance. Make this more clear in the introduction already.

RESPONSE: Based on earlier studies, summer predation by wolves on reindeer focuses on calves. These results have been now mentioned, reasoning the treatment of calf/female ratio as a dependent variable. 

4) Your analyses 2: You first tested reindeer abundance ~ moose abundance and found a strong positive relationship between the two, with and without lag. Still, you chose to include both reindeer and moose abundance as predictors when modelling a) the calf/female ratio and b) wolf abundance. The strong collinearity of reindeer and moose abundance can give faulty model results and should be avoided.

RESPONSE:

We re-analyzed data using beta regression for the calf/female ratio and GLM models together with GLS models for analysis of relationships between populations. To avoid errors due to multicollinearity, reindeer and moose abundances were not entered into the same model as independent variables.

Below is the more detailed description about data re-analysis: 

The multicollinearity of the predictors such as moose and reindeer abundances: 

As referees pointed out, the wildlife populations correlate with each other, and multicollinearity is usually a problem in the interpretation of the results. However, in our case our models are just explanatory models although we illustrated the effects using the effects plots for the predictors, fixing the other predictor(s) at their mean values. The correlating predictors, being in the same model also “purified” each other’s effects. We wanted to see how they worked when they were added to the model one by one. They “purified” together the autocorrelated errors as well in some of the models. The models for the calf/female ratio by adding the variable one by one using GLM and binomial distribution:

1. Wolves only

Coefficients:

 Estimate Std. Error t value Pr(>|t|) 

(Intercept) 0.135406 0.198021 0.684 0.502 

Wolves -0.037399 0.007529 -4.967 8.56e-05 ***

2. Wolves and moose

Coefficients:

 Estimate Std. Error t value Pr(>|t|) 

(Intercept) 0.23356 0.50458 0.463 0.649000 

Wolves -0.03759 0.00779 -4.826 0.000136 ***

Moose -0.03326 0.15651 -0.213 0.834104 

---

Signif. codes: 0 ‘***’ 0.001 ‘**’ 0.01 ‘*’ 0.05 ‘.’ 0.1 ‘ ’ 1

(Dispersion parameter for quasibinomial family taken to be 6.946016)

3. Wolves and reindeer 

Coefficients:

 Estimate Std. Error t value Pr(>|t|) 

(Intercept) -0.3114357 0.4046989 -0.770 0.451549 

Wolves -0.0331237 0.0081116 -4.084 0.000697 ***

Reindeer 0.0003449 0.0002736 1.260 0.223625 

---

Signif. codes: 0 ‘***’ 0.001 ‘**’ 0.01 ‘*’ 0.05 ‘.’ 0.1 ‘ ’ 1

(Dispersion parameter for quasibinomial family taken to be 6.311177)

All three explanatory variables (wolves,reindeer,moose)

Coefficients:

 Estimate Std. Error t value Pr(>|t|) 

(Intercept) 0.4656050 0.3680501 1.265 0.222917 

allwolves -0.0202632 0.0069474 -2.917 0.009617 ** 

REINDEER 0.0018684 0.0004542 4.114 0.000725 ***

moose.density -0.9279260 0.2464483 -3.765 0.001543 ** 

---

Signif. codes: 0 ‘***’ 0.001 ‘**’ 0.01 ‘*’ 0.05 ‘.’ 0.1 ‘ ’ 1

(Dispersion parameter for quasibinomial family taken to be 3.70951)

(Dispersion parameter for quasibinomial family taken to be 6.575754)

Thus, both populations, reindeer and moose are strongly correlated with each other. This can be seen also in Fig. 2. The significant negative coefficient of moose density is difficult to understand and interpret. Thus, the model 4 has been excluded.

Certain multicollinearity is evident in the models for wildlife populations, in general, and it might reveal some interesting features of the relations between the populations when the models were built by adding the variables one by one and testing also the interaction effects, as we have done in our modeling approach. We skipped the interaction effects in the final model versions because they were clearly non-significant.

GLS vs GLM models

In table 2 we presentd GLS models and log-transformations (if needed) for the response variables to obtain better residuals, although the data was count data. Poisson or negative binomial models would be a more reasonable alternative to the count data like this. 

The reason for our selection in our manuscript was the fact that in R the pseudo-likelihood based function glm in MASS package could not allowed autoregressive structures for the error term, but gls function in nlme package could. Especially when a certain population was modeled using just one other population as an explanatory variable, the autoregressive correlation structure was highly needed (AR1 or AR2) for reasonable standard errors for the estimated coefficients and the corresponding p-values. However, Poisson models using the AR-structures could be computed using R function glmmPQL. The function needs the specification of a random part of the model. Our data is non-hierarchical, and the only way to use glmmPQL, that can be handle the AR-errors structures is to specify a “pseudo” random part that gives the variance very near to zero. In addition, the function glmmPQL estimates so called residual, corresponding to the square-root of dispersion parameter. The only problem using that function is slightly biased degrees of freedom, because of the estimated random effect. It could be corrected using e.g. Kenward-Rogers df corrections, but as far as we know, they are not available for almmPQL (also pseudo-likelihood method).

In R, there are some other packages for Poisson or negative binomial models and autoregressive error structures (e.g. glarma), but the predictions with confidence intervals are much harder to compute, because they can not utilize the smart R package effects in the computing of the predictions with their confidence intervals.

Our solution to the referee’s comment to replace GLS using GLM was to add the corresponding GLM-based t-values and corresponding p-values to table 2 using glmmPQL function and illustrate the effects in the figures using the GLM-approach. The results and interpretations did not change considerable, but no possible negative predictions were met otherwise than using the GLS. Interpretation of the results remained the same. 

We also tested how reindeer population size and the calf/female ratio in reindeer was related to two-way interaction between wolf population size and moose abundance. 

Binomial distribution in the models for calf / female ratio vs. beta distribution 

A referee: ”… For this, beta-regression would be more adequate than a GLM with binomial family”

The calf / female ratio is merely a ratio than the proportion, as the referee mentioned. Beta distribution would be nice and alternative option to model the phenomenon.

 In fact, our binomial model would be approximately expressed as the proportion of the females with a calf (success) / the females without a calf (failure). Using the proportion of females with calves / all females weighted by all females would give the same results.

The binomial model awaked the problem of overdispersion. This problem could be avoided by using the beta distribution, but also by estimating the dispersion parameter in the binomial model, as we did in our original model versions.

In R, there is beta distribution available in package betareg.

Although the results (their interpretations) were almost the same, as we illustrate later, we used the beta distribution in our models, because it is more appropriate to interpret the calves by females as a ratio, such as the referee suggested. A good think is that the R package betareg can today utilize the R function effects (in R package effects) to obtain the predicted values with the confidence intervals!

Binomial model 

> mod<-glm(calves/females ~ wolves+reindeer+moose.density,weights=females,family="quasibinomial",data=d2)# Uusi malli uudella datalla 160519!!!

> summary(mod)

Call:

glm(formula = calves/females ~ allwolves + REINDEER + moose.density, 

 family = "quasibinomial", data = d2, weights = females)

Deviance Residuals: 

 Min 1Q Median 3Q Max 

-3.4961 -1.0399 0.2541 0.7820 2.9779 

Coefficients:

 Estimate Std. Error t value Pr(>|t|) 

(Intercept) 0.4656050 0.3680501 1.265 0.222917 

allwolves -0.0202632 0.0069474 -2.917 0.009617 ** 

REINDEER 0.0018684 0.0004542 4.114 0.000725 ***

moose.density -0.9279260 0.2464483 -3.765 0.001543 ** 

---

Signif. codes: 0 ‘***’ 0.001 ‘**’ 0.01 ‘*’ 0.05 ‘.’ 0.1 ‘ ’ 1

(Dispersion parameter for quasibinomial family taken to be 3.70951)

 Null deviance: 288.820 on 20 degrees of freedom

Residual deviance: 62.182 on 17 degrees of freedom

AIC: NA

Number of Fisher Scoring iterations: 4

Beta regression model with logit-link

# vasaos denotes the ratio of calves/females

> mod <- betareg(vasaos ~ allwolves+REINDEER+moose.density,link="logit", data = d2)

> summary(mod)

Call:

betareg(formula = vasaos ~ allwolves + REINDEER + moose.density, data = d2, link = "logit")

Standardized weighted residuals 2:

 Min 1Q Median 3Q Max 

-2.4928 -0.6743 0.1543 0.5653 2.5688 

Coefficients (mean model with logit link):

 Estimate Std. Error z value Pr(>|z|) 

(Intercept) 0.290914 0.338926 0.858 0.390705 

allwolves -0.022109 0.006349 -3.482 0.000497 ***

REINDEER 0.001475 0.000464 3.178 0.001481 ** 

moose.density -0.693933 0.253986 -2.732 0.006292 ** 

Phi coefficients (precision model with identity link):

 Estimate Std. Error z value Pr(>|z|) 

(phi) 72.21 22.15 3.26 0.00111 **

---

Signif. codes: 0 '***' 0.001 '**' 0.01 '*' 0.05 '.' 0.1 ' ' 1 

Type of estimator: ML (maximum likelihood)

Log-likelihood: 31.56 on 5 Df

Pseudo R-squared: 0.7031

Number of iterations: 46 (BFGS) + 4 (Fisher scoring)

Results are not completely the same, but the interpretations are. The example here was the model with the highly correlating population of moose density and reindeer abundance. The response, vasaos in the beta model denotes the proportion of calves by females.

Comparing the alternative models using pseudo-R2

The R2 values were missing in table 2. In our corrected version we computed the R2 values for the GLS models. They are pseudo-R2 values which cannot be interpreted as the proportions explained by the predictors of the models, merely just for the comparisons between the GLS models, similarly to the models in Table 1 where the R2 values represented the pseudo-R2 for the quasi-likelihood GLMs.

We used R-package rcompanion in the computation of the pseudo-R2 values for GLS. The function nagelkerke in the package gives three different pseudo-R2 values. We used Cox and Snell-values. The other values usually gave rather similar interpretation. In some of the models it might be a little confusing that the values are slightly negative. It happened when the p-value of the only predictor was very or rather near to 1. 

5) Your results and discussion: Your main results are 1) negative relationship for calf/female ~ wolf abundance, which might indicate calf predation by wolves; 2) a negative relationship for wolf and positive for moose for reindeer abundance ~ wolf abundance + moose abundance; 3) No relationship between wolf abundance ~ moose abundance. Although the latter two findings give some support to APH, but not to ACH, you still put a lot of emphasis on ACH in the discussion, and you even propose reduction of moose population size as a measure to decrease predation on reindeer. I think evidence for this statement is not present in your results.

RESPONSE: We had highlighted our main findings in Results and Discussion as suggested by the referee. We notify that our results be more supportive on APH than ACH but cautiously because we considered just correlations between populations. In the first chapters of discussion we briefly go through the primary findings and tell that our results are probably more supportive to APH than ACH hypothesis. 

6) The English needs to be improved. I have not focused on he language in my review, but can do so in a revised manuscript.

RESPONSE: The English has been checked by a native but may need further improvement. 

Specific comments:

L1: The title is misleading. You did not study predation. You assume that predation may be an underlying mechanism of the observed correlations between the time series, that’s all.

RESPONSE: We changed the title to match better with the content. 

L10-13: rewrite this first paragraph of the abstract. Include the alternative prey hypothesis and theory of apparent competition, end up with your predictions – see general comments

RESPONSE: Done 

L39-40: the message of this sentence is unclear as it is written know. You refer to the paper of Holt and Lawton (1994) and took some of their wording in the abstract into your sentence. They write “…Theory suggests victim-species coexistence depends on particular conditions…”. By using the same wording, the context went missing. I think you can delete this first sentence and rewrite the entire first paragraph of the introduction. Explain the two hypotheses that describe direct and indirect interactions of prey species with a shared predator.

RESPONSE: Now as follows: Background. The alternative prey hypothesis describes a mechanism where that the population size of secondary prey species, endangered wild forest reindeer (Rangifer tarandus fennicus) in our study, may decrease whenever abundance of the principal prey, moose (Alces alces) goes down through dietary shift by a generalist predator, wolf (Canis lupus), from moose to reindeer. Apparent competition is a process where the population size of reindeer can be assumed to decrease as a result of the increased population size of wolf which results from increasing prey biomass that is highly related to moose abundance in our study area in Finland. 

L41: Apparent competition – is it a theory or a hypothesis? I see both is used in literature. I personally would prefer hypothesis, so it is on the same line as the alternative prey hypothesis.

RESPONSE: Now both the apparent competition and the alternative prey hypotheses are termed as hypotheses.

L42: I would put APH before ACH, because APH describes a mechanism (prey switching), while ACH is more of a process. This would also change the order in the text of the following lines.

RESPONSE: The order changed.

L68-73: It would make sense to first talk about predator control, so move the last sentence up, and so talk about the regulation of the main prey, and so the combination of both.

RESPONSE: The order changed.

L86: The reproductive rate is so much more than calf/female ratio. Make it more specific and introduce the reader to the wolf being a calf specialist – if this is true.

RESPONSE: We removed the term ‘reproductive rate’ from the ms. We use now only calf/female ratio.

L103: Any chance to incorporate bear data into your models? I think they might be an important predator on neonate calves. What about lynx?

RESPONSE: We do not have proper data about changes in bear and lynx population. We quoted to our unpublished small data about bear predation; we had tracked two bears during early summer and found only moose calves at kill sites. It may be that the very scattered distribution of female forest reindeer during calving time decreases bears’ motivation to actively seek for the calf reindeer that are small compared to moose calves. 

L124: Why had collared animals to be snow-tracked? To find pack size? Clarify!

RESPONSE: Yes, clarified.

L126-127: Clarify!

RESPONSE: Described with more details: The moose abundance index based on records by hunters. Moose hunting clubs are operating within their own territory of approximately 5 000 ha. Each hunting club records the number of moose observations per hunting day during the autumn hunting season (from the last Saturday of September to the end of December) and the resultant index provides the annual mean number of moose per hunting day for our study area. We used this index because moose density was not estimated for years 1996-1999. 

L132: So the reindeer counts were a sample, not a full count? Or is the number of observations identical with abundance?

RESPONSE: Yes, reindeer counts were full counts. Clarified.

L133: Number of observations: Is this the number of females, the number of animals, or the number of days people have been out observing?

RESPONSE: This means the number of females. We have clarified this.

L138: I guess this needs to be dependent, not independent.

RESPONSE: Yes. Changed.

L160 onwards: GLM with proportional data, GLS and AR-structures are standard methods, and there is no need to show the model equations. However, the calf/female ratio is not really a proportion, it is a ratio. For this, beta-regression would be more adequate than a GLM with binomial family.

Results: I would move the abundance sub-chapter (relationships between species-abundances) up, so it comes right after the first paragraph on trends.

RESPONSE: Changed as the referee proposes. The calves/females are treated as a ratio in beta-regression. The order has been changed in all sections. 

L243: Where do you present those pseudo R-squared?

RESPONSE: They are now presented in Table 1 and 2. 

L248: See general comment on collinear predictors, so I don’t think you can include both reindeer and moose. But anyhow: What do you define as significant? In Table 2, p-values of those two predictors are > 0.05. 

RESPONSE: True, changed, only the p values <0.05 have now been mentioned as significant.

L252: No, you did not study predation on boral wild reindeer! See general comment 1).

RESPONSE: This sentence has been omitted. 

L255: …major reason…? Rather: …one of the reasons…

RESPONSE: changed.

L261-264: But the wolf population did not really decrease after moose population decreased. Which brings to my mind that the ACH should test predator numerical response not only as a function of its principal prey, but rather of the total prey base, i.e. the sum of available prey biomass irrespective of species. You however use an index for moose abundance, so it will be difficult to estimate the sum of available biomass of reindeer and moose.

RESPONSE: We mention that moose density is the main determinant of prey biomass and therefore moose abundance is relatively relevant parameter when considering ACH.

L288: I assume the gravel roads are a consequence of forestry. Clarify, maybe call them forest roads. Has their density changed throughout the years? Or has their use changed?

RESPONSE: Yes. This is now changed. The boom in building forest roads was in 1970s and 1980s. Their density has increased just marginally during our study period. This is notified in text.

L292-293: This is a really strong statement that has no direct support from your data, see also general comments 1) and 5).

RESPONSE: True, a sentence has been omitted. 

L314: Moose abundance was positively related with reindeer abundance, and I consider the model looking into calf/female ratio ~moose abundance + reindeer abundance as invalid due to collinearity of the predictors. So no support for a negative relationship between moose abundance and “predation risk”.

RESPONSE: Good point, the sentence modified into as follows: The reduction in the abundance of primary prey might (seems) decrease the predation risk although this was not supported in our study. 

L525 and else in the text: You use abundance, population, population size and density (e.g. Figure 5b) intermittently. Be consistent.

RESPONSE: Changed to be consistent.

Figure 1: Finland is not labelled, while all other countries are! The map allows for inclusion of more information, which could make the study easier to understand. You could for example add wolf territories, e.g. as overlapping territories from all years. And/or the network of forest roads. Maybe also the area of the other wild reindeer population. Anything to make it more informative!

RESPONSE: A new map has been drawn. The Fig 1. shows Finland as labelled, wolf territory boundaries and road network.

Figure 2: It would make sense to include two similar graphs for calf/female ratio.

RESPONSE: Calf/female ratio is now shown in Fig. 2. Fig. 3 omitted.

Figure 5: The y-axis goes below zero, which does not make sense for wolf abundance. This makes me think that you maybe rather use a Poisson-regression when comparing wolf abundance (a count) to moose abundance index (or reindeer abundance).

RESPONSE: Changed as the referee suggests.

Reviewer #2: The authors have been studying wolf predation on wild forest reindeer in a system where an endangered predator (wolf) has a primary non-endangered prey species (moose) and secondary endangered prey species (wild forest reindeer). In the manuscript the authors have used correlative approach to investigate species abundance interactions on reindeer and wolf population sizes. In addition, they have also used the same approach to investigate calf/female ratio in winter herds of wild forest reindeer. In former studies this ratio has been shown to be related on wolf abundance and in this manuscript the authors have broadened their former study by analysing longer data period and adding the yearly abundance of moose into their models. Their main findings were that the calf/female ratio was negatively associated negatively to both wolf population size and moose abundance. Wild forest reindeer size was only dependent on wolf population size in model where the moose abundance was entered as another independent variable. There was not strong evidence on effect of moose abundance or reindeer population size on wolf population size.

The long term time series for wildlife species used in the study are imposing and the statistical analyses are rather comprehensive, even though there have been some problems in the analyses because of autocorrelations. The study system and the results are very interesting. However, the interpretation of the causal effects behind the species population size variation are difficult because of the correlative approach used in the study. The fact that moose population size is heavily regulated via hunting and wolf population is fluctuating mainly because of poaching makes interpretation of the results even more difficult. However, the observation that the predation by wolves on reindeer might be influenced by moose abundance could still have substantial management implications and the authors are discussing praiseworthy on possible management actions.

The paper is generally well written. However, especially in the abstract the authors should present their results more comprehensively to make the message of the paper clear (see below). And, please, do not use “population” as a substitute of “population size”. 

RESPONSE: We have re-analyzed data to avoid errors due to multicollinearity. The term ‘population’ has been changed into ‘population change’ when the sentence does not clearly indicate that there is a question of population size. 

Minor points:

Abstract:

Line 15: “…reproductive rate…”. What does this mean? Is it your calf/female ratio. Reproduction is generally measured by gross reproduction rates or net reproduction rates that generally indicate the ratio between the sizes of the daughter's and mother's generations and you are not really measuring them.

RESPONSE: Each ‘reproductive rate’ changed to ‘calf/female’ ratio. 

Line 15: “wild reindeer”. Sometimes wild reindeer and wild forest reindeer. Please, be consistent.

RESPONSE: Now only “wild forest reindeer”, because it is an official name of the subspecies.

Line 20: “Reindeer and moose abundances were highly correlated…”. Is this already a result and should be in your Results section of your abstract. You are actually repeating this result also on line 

24-25. “The trends in reindeer population size and moose abundance were almost identical”. 

RESPONSE: Changed like this. Repeats removed. 

Line 27: “Change in reindeer population between consecutive winters”. Should not this be “Change in reindeer population *size* between consecutive winters”.

RESPONSE: Yes, changed.

Line 28: “The calf/female ratio was closely related to wolf population size”. The calf/female ratio was closely related *negatively* to wolf population size.

RESPONSE: Changed.

Line 29: “the reindeer population was related to the wolf population”. I suppose that you mean “the reindeer population *size* was related *negatively” to the wolf population *size*. Other wise this text sounds funny; and tell also that the relationship is negative. …”

RESPONSE: Corrected as the referee suggests.

Line 30: “The wolf population was…”. Should be “The wolf population *size* was

RESPONSE: Yes. Changed. 

Introduction: “

Lines 85-90: I suggest that the authors construct a table where they show what are the predictions of their two theoretical models and how their different results support or do not support the two hypotheses. 

RESPONSE: In this reviewed version we now express the hypotheses and predictions in much more coincided and clearer fashion in the Abstract, Introduction and Discussion. We think that these changes a bit decrease the need for a table, but if it is seen necessary we are ready to instruct such into the manuscript. 

Data:

Lines 132-133: “the calf/female ratio was weighed against the annual number of observations”. In the analyses?

RESPONSE: In the present statistical analysis (betareg) we use calf/female ratio. 

Lines 145-146: “The aim of the study was to model the relationships between these populations”. Or: “The aim of the study was to model the interactions in abundances of these populations” or did you really model the relationship between the populations?

RESPONSE: Changed into “

Line 147-148: “In addition, the relationship of the calf/female ratio to the wolf population..” Or: “In addition, the relationship of the calf/female ratio to the wolf population *size*”.

RESPONSE: Correct. Changed.

Result:

Line 217: “Reindeer population and moose abundance.”Or “ Reindeer population *size* and moose abundance “

Line 221: “The annual growth rate of the wild forest reindeer population was related to the number of calves weighted by the number of females (the calves/females ratio) in the linear model Y = -0.16 + 0.473 * X, t = 2.57, p = 0.020).” Is the number of calves weighted by the number of females” really correct here. Thus what are your Y and X here. Please, clarify.

RESPONSE: Y and X added in into the sentence.

Line 225: “but the ratio was related to the wolf population” …. Should be: “but the ratio was related to the wolf population *size*”.

RESPONSE. Yes, changed. 

Line 227-228: “the calf/female ratio was related to the wolf population in a highly 28 significant fashion”. Should be: “the calf/female ratio was related to the wolf population *size*” in a highly significant fashion”.

RESPONSE: YES, changed

Line 228-229: “while no relationship between the ratio and the reindeer population *size* existed”. However, there was significant association between reindeer population size and calf/female ratio in your model with three independent variables. Should you mention it, too?

RESPONSE: We have omitted the results based on three independent variables to avoid errors due to multicollinearity because reindeer and moose abundances were so highly correlated, as suggested by the referee 1.

Line 237-241: “but in a model where wolf and moose population *sizes* were entered as independent variables, reindeer population size was negatively related to the wolf population *size* and positively related to the moose population *size*. In models with a one-year lag, the reindeer population *size* was not related to the wolf population *size* alone but was related to both the wolf population *size* and moose*abundance* in a model where both were entered as independent variables”.

RESPONSE: Changed.

Lines 247-250: “Wolf. In a model without a lag, the wolf population size was positively related to moose abundance and negatively related to reindeer population size (Table 2, Fig. 5). In a model where 249 the wolf population size was the dependent variable with a one-year lag, neither the moose nor reindeer population *size* was significantly related to the wolf population (Table 2).

RESPONSE: Changed.

Discussion:

Lines 254-255: “We found that calf/female ratio in reindeer was negatively related to wolf population *size*…

Lines 260-261: “The stabilization of the reindeer population size at the same time the moose population *abundance* decreased…”

Figures:

Fig 4: “Relationship of the calf/female ratio of wild forest reindeer to the wolf population *size*in…”

Fig 5: “Relationships of reindeer population *size*(a) and moose abundance (b) to wolf population *size*…”

RESPONSE: “Population” is now termed as “population size”.

---

## [Editor Report · Decision Letter 1]

18 Oct 2021

Calf/female ratio and population dynamics of wild forest reindeer in relation to wolf and moose abundances in a managed European ecosystem

PONE-D-21-02662R1

Dear Dr. Kojola,

We’re pleased to inform you that your manuscript has been judged scientifically suitable for publication and will be formally accepted for publication once it meets all outstanding technical requirements.

Kind regards,

Lalit Kumar Sharma

Academic Editor

PLOS ONE
---

## [Editor Report · Acceptance letter]

3 Dec 2021

PONE-D-21-02662R1 

Calf/female ratio and population dynamics of wild forest reindeer in relation to wolf and moose abundances in a managed European ecosystem 

Dear Dr. Kojola:

I'm pleased to inform you that your manuscript has been deemed suitable for publication in PLOS ONE. Congratulations! Your manuscript is now with our production department. 

Kind regards, 

on behalf of

Dr. Lalit Kumar Sharma 

Academic Editor

PLOS ONE